# CG>TG mutation frequency as negative predictor of homologous recombination deficiency in ovarian and breast cancer
Eva Romanovsky [1], Michael Menzel [1], Klaus Kluck [1], Susanne Beck [1], Markus Ball [1], Peter Schirmacher[1,2], Daniel Kazdal [1,2], Albrecht Stenzinger [1,2,3] & Jan Budczies [1,2,3] ✉

Homologous recombination deficiency (HRD) is a predictive biomarker for PARP inhibition and platinum-based chemotherapy. While copy number alteration-based scores such as $HRDsum = LST + TAI + LOH$ are included in therapy approvals, single base substitutions (SBS) are underinvestigated as predictors of HRD. WES data of the TCGA pan-cancer cohort and an in-house ovarian cancer cohort were annotated by alterations in *BRCA1/2* and additional genes causative of HRD. Using this reference, the new biomarker $f_{deam}$ defined as frequency of C > T transitions at CpG sites in relation to all SBS and HRDsum were compared for the detection of HRD. In the TCGA ovarian cancer, the in-house, and the TCGA breast cancer cohorts, $f_{deam}$ performed non-inferior to HRDsum (AUC = 0.84, AUC = 0.85, and AUC = 0.88). The cutpoint $f_{deam} = 13.1\%$ maximized the balanced accuracy in the TCGA ovarian cancer cohort and resulted in sensitivity = 89% and specificity = 77% in the in-house cohort. In a simulation study, $f_{deam}$ retained high sensitivity for HRD detection and outperformed HRDsum in tumors of purity 40%, 20%, and 10%. Overcoming the limited robustness against low tumor purity, the new biomarker can contribute to a more sensitive detection of HRD in clinical samples. Further studies are warranted to confirm its clinical validity and utility and explore its potential for liquid biopsies.

Defective DNA repair causing genomic instability is a hallmark of cancer[1,2]. While human cells possess a complex repertoire of interlocked mechanisms to maintain genomic integrity, homologous recombination (HR) is the only pathway capable of repairing DNA double-stand breaks in a flawless manner[3]. Malfunction of this pathway, termed homologous recombination deficiency (HRD), is associated with increased sensitivity to platinum-based chemotherapy and to PARP inhibitors (PARPi) by synthetic lethality[4]. Biomarkers of HRD defined either by the detection of deleterious mutations in *BRCA1/2* or other HR genes ("cause") or analysis of genomic instability ("effect") are included in approvals and clinical guidelines of PARPi in ovarian, breast, prostate, and pancreatic cancer[5].

Recent ESMO recommendations have ranked HRD tests for clinical validity (accuracy of prediction of PARPi sensitivity) and clinical utility (accuracy of prediction of PARPi benefit) in ovarian cancer[6]. Among the diagnostic tools testing the cause, clinical validity and utility of *BRCA1/2* germline and of *BRCA1/2* somatic mutations were ranked as "good" in both the 1st line and in the relapsed setting, while they were ranked as "marginal" or "absent" for mutations in other HR genes. Among the diagnostic tools

testing the effect, the recommendations include a score for loss of heterozygosity (LOH) the sum score $HRDsum = LOH + LST + TAI$ that additionally assesses large-scale transitions (LST) and telomeric imbalance (TAI). These scores are calculated from genome-wide copy number alterations (CNAs) detected by genotyping or NGS[7–10]. Clinical validity and utility of HRDsum were ranked as "good" in both settings, while the clinical validity of LOH was ranked as "good" and its clinical utility was ranked as "absent" and "good" in the 1st line and the relapsed setting, respectively.

Despite the demonstrated clinical validity and utility, the presently available HRD tests suffer from the following limitations: (1) The main limitation of *BRCA1/2* mutation testing is to miss HR-deficient tumors with mutations in other HR genes, methylation of *BRCA1* or other HR genes, and other HRD-causing alterations such as gene expression silencing[11–13]. (2) Limitations of genomic instability scores (GIS) include insensitivity for detecting restored proficient HR repair that e.g., caused by reversal mutations[14–16] and the difficulty of cutpoint optimization needed to define a clear-cut separation of HR-deficient from HR-proficient tumors. Recent data support the view that a one-fit-all cutpoint

[1]Institute of Pathology, Heidelberg University Hospital, Heidelberg, Germany. [2]Center for Personalized Medicine (ZPM), Heidelberg, Germany. [3]These authors contributed equally: Albrecht Stenzinger, Jan Budczies. ✉e-mail: jan.budczies@med.uni-heidelberg.de

for the HRDsum is unlikely to be optimal and cutpoints should rather be defined in a tumor-type-specific manner[13]. (3) Low tumor purity is a clinically relevant confounder for the detection of HRD using copy number-based GIS such as HRDsum[13,17]. While the sensitivity of HRD detection is substantially reduced for tissue samples with tumor purity ≤40%[13,18], the selection of samples with high tumor purity can be difficult or impossible when dealing with clinical samples.

The aim of the study was to derive a predictor of HRD from the spectrum of SBS. To this end, we analyzed WES data of the pan-cancer TCGA cohort and of an in-house ovarian carcinoma cohort. An earlier established classification system based on mutations *BRAC1*, *BRCA2*, and other HR-genes[13] was employed to generate a ground truth for HRD status. Mutational signatures SBS3 and SB8 that were fitted by the commonly used tools SigProfiler[19] and FitMS[20] significantly correlated with HRD but failed to separate HR-deficient form HR-proficient tumors with high accuracy. Comparative analysis of the frequency of 96 mutations types in HR-deficient and -proficient tumors motivated the introduction of a new biomarker $f_{deam}$ defined as frequency of C > T transitions at CpG sites in relation to all SBS. The new biomarker was non-inferior to HRDsum in the detection of HRD in ovarian and breast cancer and more robust against low tumor purity compared with HRDsum.

## Results

We sought to evaluate different types of GIS to distinguish between HR-deficient and -proficient tumors. To this end and to allow for an unbiased comparison between different GIS, we classified tumors according to the underlying genetic and epigenetic alterations causing HRD using the earlier developed stratification in the classes H1a, H1b, H2a, H2b, and H3[13]. As the main benchmark, we analyzed the ability of GIS to separate the tumors with deleterious mutations in *BRAC1/2* affecting all alleles or with *BRCA1* promoter hypermethylation (class H1a*) from tumors without genetic alternations in the HR pathway and without *BRCA1* promoter hypermethylation (class H3). The performance of GIS was evaluated in the ovarian cancers of the TCGA cohort (TCGA-OV), the breast cancers of the TCGA-cohort (TCGA-BRCA), the remaining tumors of the TCGA excluding ovarian and breast cancer (TCGA-PANCAN*), and a cohort of 231 ovarian cancers diagnosed at the Heidelberg Institute of Pathology (HD-OV).

### Mutational signatures

We evaluated SBS signatures and two methods for fitting (SigProfiler and FitMS) for the ability to separate HR-deficient and –proficient tumors. In TCGA-OV, the signatures SBS1, SBS2, SBS3, SBS5, SBS8, SBS13, and SBS18 were detected in at least 12 tumors and analyzed further (Fig. 1A, B, Supplementary Fig. 1A, B). Using SigProfiler, the levels of SBS3, SBS8, and SBS13 correlated significantly with the HRD status. Using FitMS, the levels of the signatures SBS1, SBS2, SBS3, SBS5, and SBS8 correlated significantly with the HRD status. Both methods agreed on a significant correlation of the levels of SBS3 and SBS8 with HRD status. Analyzing the area under the receiver operating characteristic curve (AUC), for both SBS3 and SBS8, FitMS outperformed SigProfiler in the separation of HR-deficient and -proficient tumors ($p = 0.0085$ and $p = 7.7e−05$). By contrast, for both SigProfiler and FitMS, the performance of SBS3 and SBS8 did not significantly differ (*p*-values not shown). Consistently, in the in-house cohort, HD-OV, SBS3 levels correlated with HRD status in both methods, and FitMS outperformed SigProfiler (Fig. 1C, D, Supplementary Fig. 1C, D). In TCGA-BRCA, the signatures SBS1, SBS2, SBS3, SBS5, SBS8, SBS13, and SBS18 were detected in at least 16 tumors and analyzed further (Fig. 1E, F, Supplementary Fig. 1E, F). Using SigProfiler, the levels of SBS1, SBS2, SBS3, SBS5, SBS8, and SBS13 correlated significantly with the HRD status. Using FitMS, the levels of the signatures SBS2, SBS3, SBS8, SBS13, and SBS18 correlated significantly with the HRD status. We also investigated alternative MutSigs by using a catalog of ovarian cancer-specific signatures and a catalog including signatures corresponding technical artefacts as reference and found similar results (Supplementary Fig. 2). Altogether, we observed a

significant but limited ability of SBS3 and SBS8 derived from WES data to separate HR-deficient and -proficient tumors for ovarian cancer and for breast cancer.

### Analysis of the 96 mutation types

SBS mutational signatures are based on the segmentation of the mutations in 96 mutation types defined by the substitution type, the base before the substitution, and the base after the substitution. We analyzed the levels of the 96 mutation types for being higher or lower in HR-deficient compared with HR-proficient tumors (Fig. 2). Before entering the analysis, the counts of the 96 mutation types were normalized by the total SBS load of the tumor. In TCGA-OV, 39 mutation types were significantly higher, and 6 mutation types were significantly lower in HR-deficient tumors compared to HR-proficient tumors, and 33 (84.6%) and 4 (66.7%) of these associations could be validated in HD-OV. In ovarian cancer, the by far largest effect sizes were observed for the mutation types ACG > ATG, CCG > CTG, GCG > GTG, and TCG > TTG, which were depleted in HR-deficient tumors. These mutations are all C > T transitions at CpG sites and - as CpG sites are prone to methylation - can be putatively attributed to spontaneous deamination of methylcytosine. In breast cancer, additional to those transitions of the type TCA > TTA had a similar effect size. In the pan-cancer cohort, excluding ovarian and breast cancer, ACG > ATG and GCG > GTG mutations, both C > T transitions at CpG sites, were most strongly depleted in HR-deficient tumors.

### Mutations attributed to spontaneous deamination

Based on these results, we defined a new biomarker $f_{deam}$ representing the proportion of SBSs attributed to spontaneous deamination (mutations of type CG > TG) to all SBSs detected in a tumor. We analyzed the new biomarker for correlation with HRDsum and the ability to separate HR-deficient and -proficient tumors (Fig. 3, Supplementary Fig. 3A, B). We detected significant negative correlations between $f_{deam}$ and HRDsum in TCGA-OV, HD-OV, TCGA-BRCA, and TCGA-PANCAN* (Spearman R = −0.58, −0.72, −0.3, and −0.17). Low levels of $f_{deam}$ indicated a higher probability for HRD. The new biomarker had a strong ability to separate between HR-deficient and HR-proficient tumors in TCGA-OV, HD-OV, and TCGA-BRCA (AUC = 0.84, 0.85, and 0.88), but a lower ability to achieve this separation in TCGA-PANCAN* (AUC = 0.62). In TCGA-OV, HD-OV, and TCGA-BRCA, the separation achieved with $f_{deam}$ was not significantly different from the separation achieved with HRDsum. In TCGA-PANCAN* the separation achieved with $f_{deam}$ was significantly worse than the separation achieved with HRDsum. The optimal cutpoint for separating HR-deficient and -proficient tumors was $f_{deam} = 13.1\%$ obtained by maximizing the balanced accuracy in TCGA-OV. Using this cutpoint, the sensitivity and specificity for detecting HRD-causing alteration in ovarian cancer were 87% and 79% in the training cohort (TCGA-OV) and 89% and 77% in the independent validation cohort (HD-OV).

As additional validation of $f_{deam}$ for the detection of HRD in ovarian cancer, we analyzed the independent cohort CPTAC-OV (Supplementary Fig. 4). Again, we detected a strong and significant correlation between $f_{deam}$ and HRDsum (Spearman R = −0.54). Also, the mutation-based biomarker was non-inferior to HRDsum to separate between HR deficient and HR-proficient tumors (AUC = 0.79 vs. AUC = 0.65, $p = 0.39$). In summary, $f_{deam}$ was non-inferior to HRDsum for the detection of HRD in three cohorts of ovarian cancer and one cohort of breast cancer.

### Pan-cancer analysis of the new biomarker

Pan-cancer analysis of $f_{deam}$ revealed that 19 (54%) cancer types included 25% or more tumors below the cutpoint, while for the remaining 16 (46%) cancer types more 75% of the tumors were above the cutpoint (Fig. 4A). The former group of cancer types included ovarian and triple-negative breast cancer, but also cancers for which exposure to tobacco smoking or UV light are among the known risk factors such as lung adenocarcinoma and squamous cell carcinoma (LUAD and LUSC), liver hepatocellular carcinoma (LIHC), head and neck squamous cell carcinoma (HNSC), and

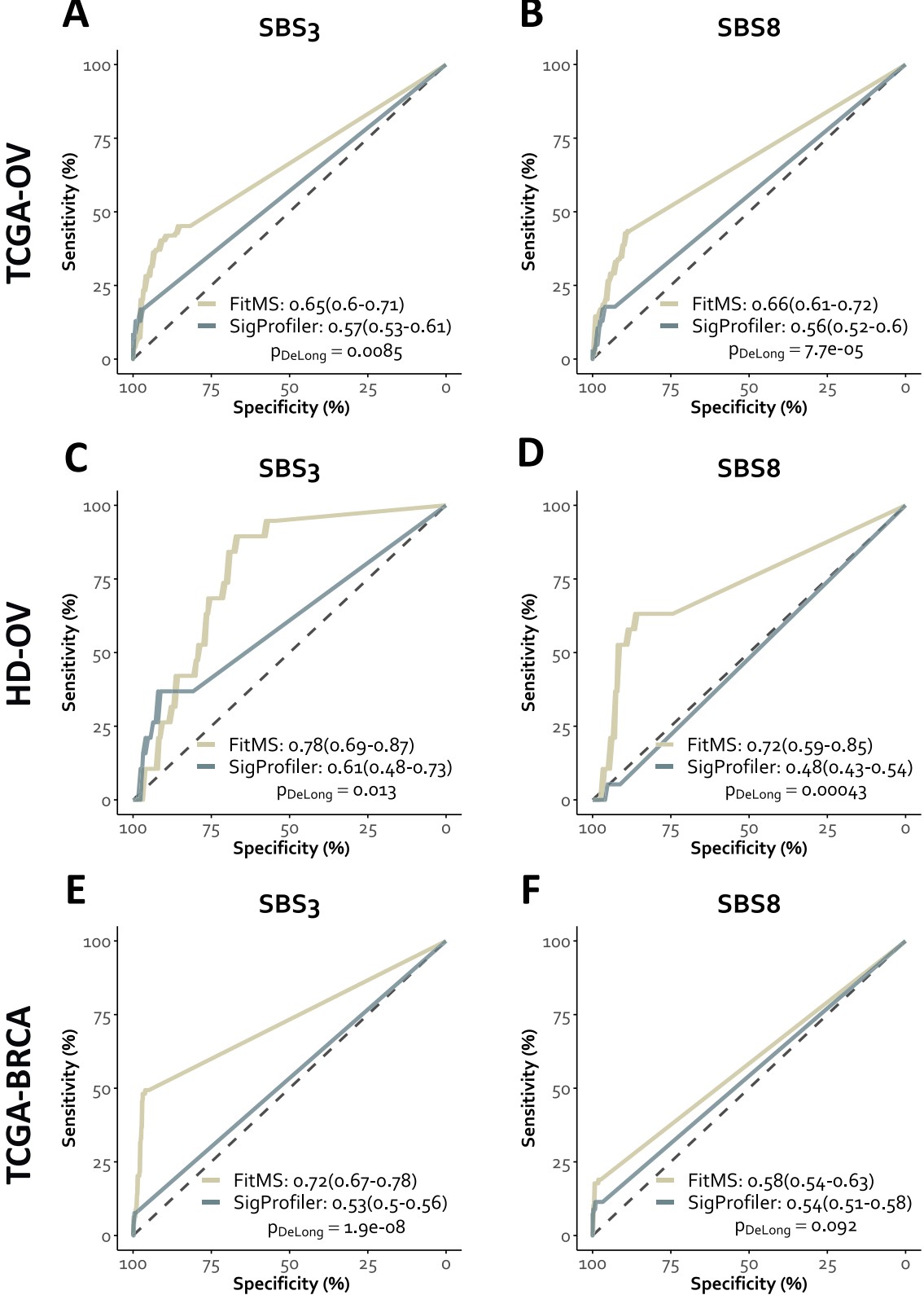

**Fig. 1 | Ability of SBS mutational signatures to detect HRD in ovarian cancer and breast cancer.** Analysis of the MutSigs SBS3 and SBS8 for the ability to separate between HR-deficient (class H1a[*]) and HR-proficient (class H3) tumors. **A**, **B** Separation of HR-deficient and -proficient ovarian carcinoma by SBS3 and SBS8 (TCGA-OV). **C**, **D** Separation of in-house HR-deficient and -proficient ovarian carcinoma by SBS3 and SBS8 (HD-OV). **E**, **F** Separation of HR-deficient and -proficient breast carcinoma by SBS3 and SBS8 (TCGA-BRCA).

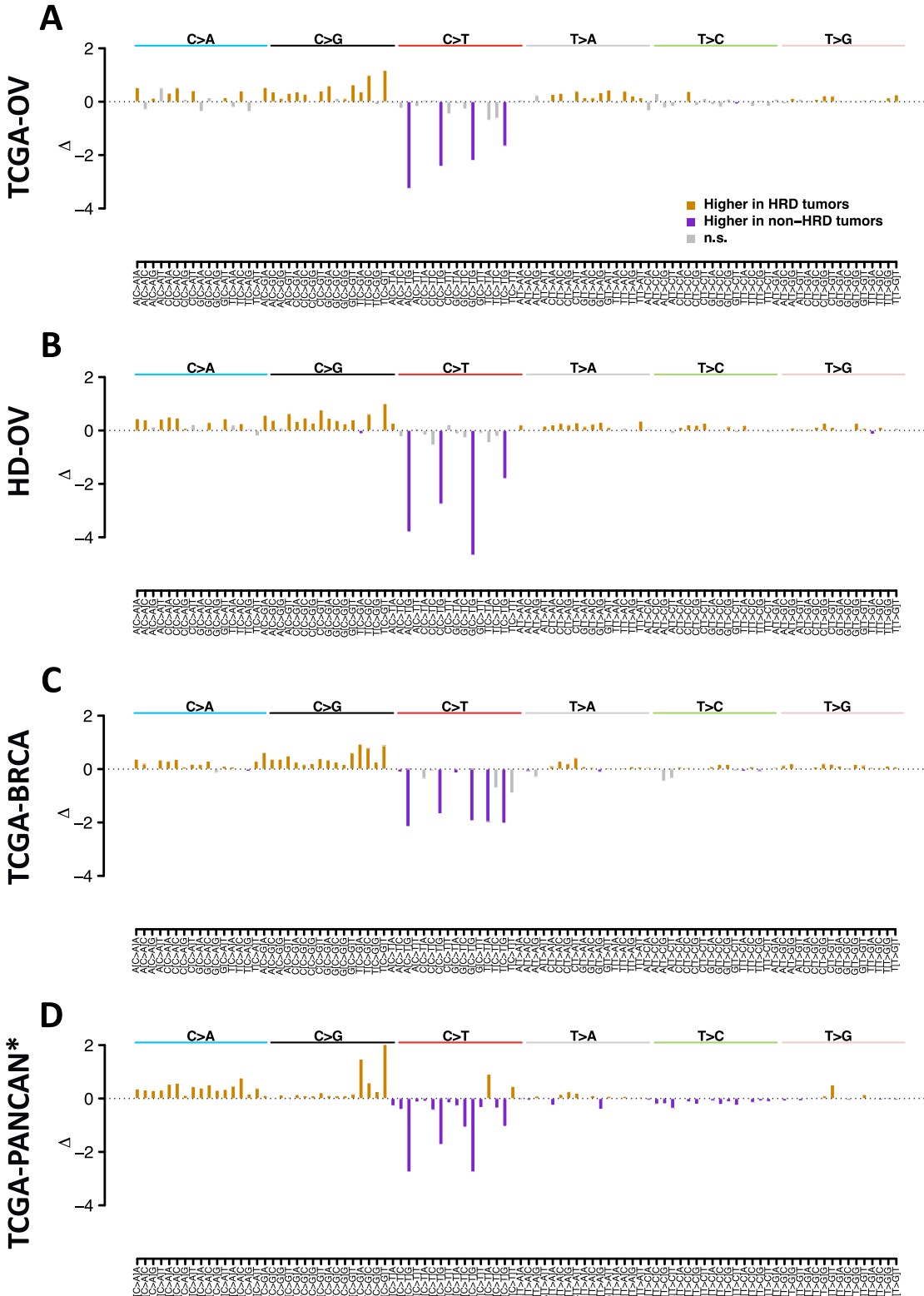

**Fig. 2 | Comparison of mutation profiles including 96 mutation types between HRD-deficient and -proficient tumors. A** Ovarian cancer (TCGA-OV). **B** Ovarian cancer, in-house cohort (HD-OV). **C** Breast cancer (TCGA-BRCA). **D** Pan-cancer excluding ovarian and breast cancer. TCGA-PANCAN* = all entities included in the TCGA except BRCA and OV.

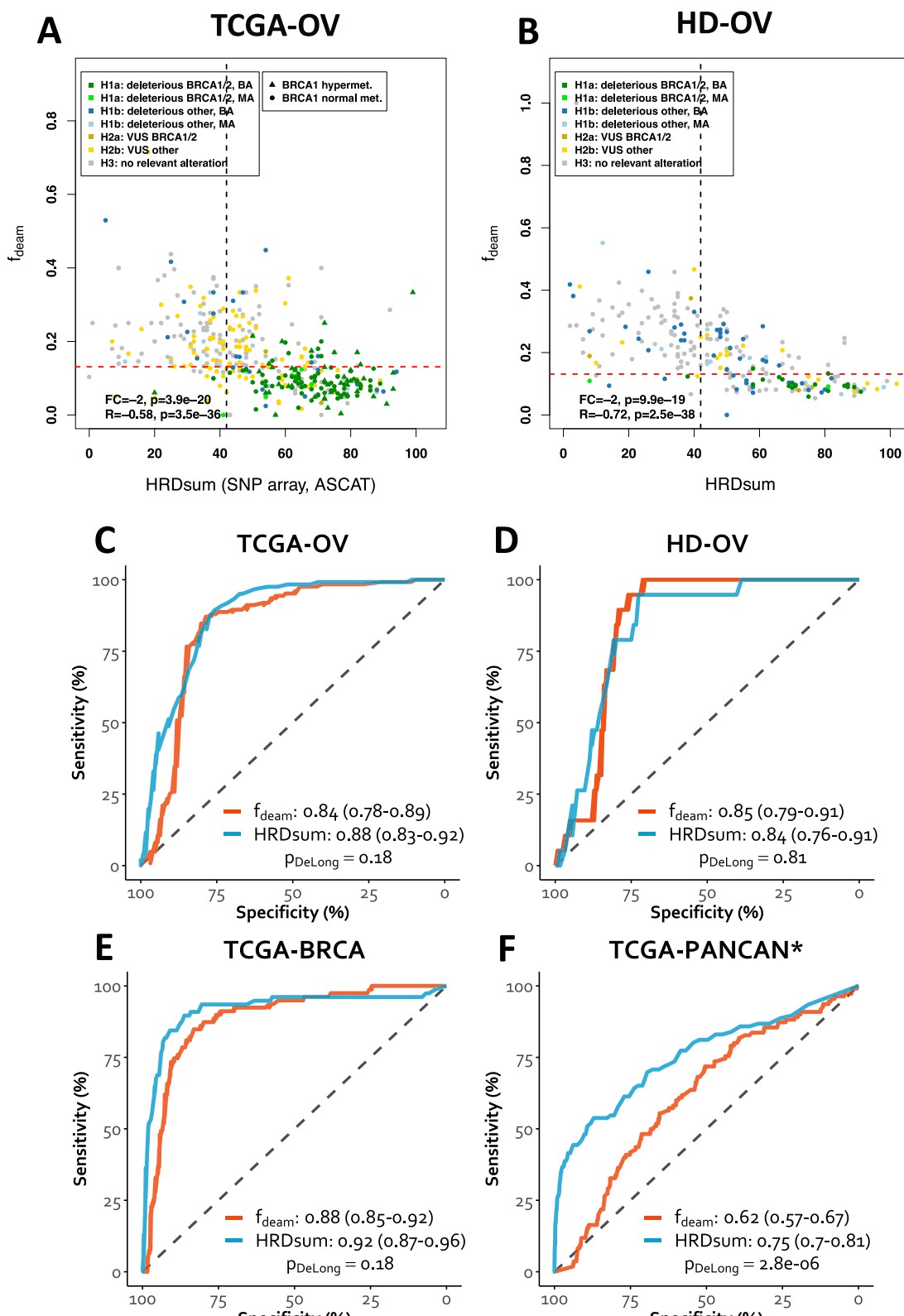

**Fig. 3 | Mutations attributed to spontaneous deamination of methylcytosine as biomarker for HRD detection.** The fraction of C > T transitions at CpG sites relative to all SBSs ($f_{deam}$) was investigated for the ability to separate between HR-deficient and -proficient tumors. **A** Correlation analysis of $f_{deam}$ and HRDsum in ovarian cancer (TCGA-OV). **B** Same as (**A**), but in the in-house cohort (HD-OV). **C** Separation between HR-deficient and -proficient ovarian carcinomas (TCGA-OV). **D** Same as (**C**), but for the in-house cohort (HD-OV). **E** Separation between HR-deficient and –proficient breast cancers (TCGA-BRCA). **F** Separation between HR-deficient and –proficient pan-cancer excluding ovarian and breast cancer. TCGA-PANCAN[*] = all entities included in the TCGA except BRCA and OV.

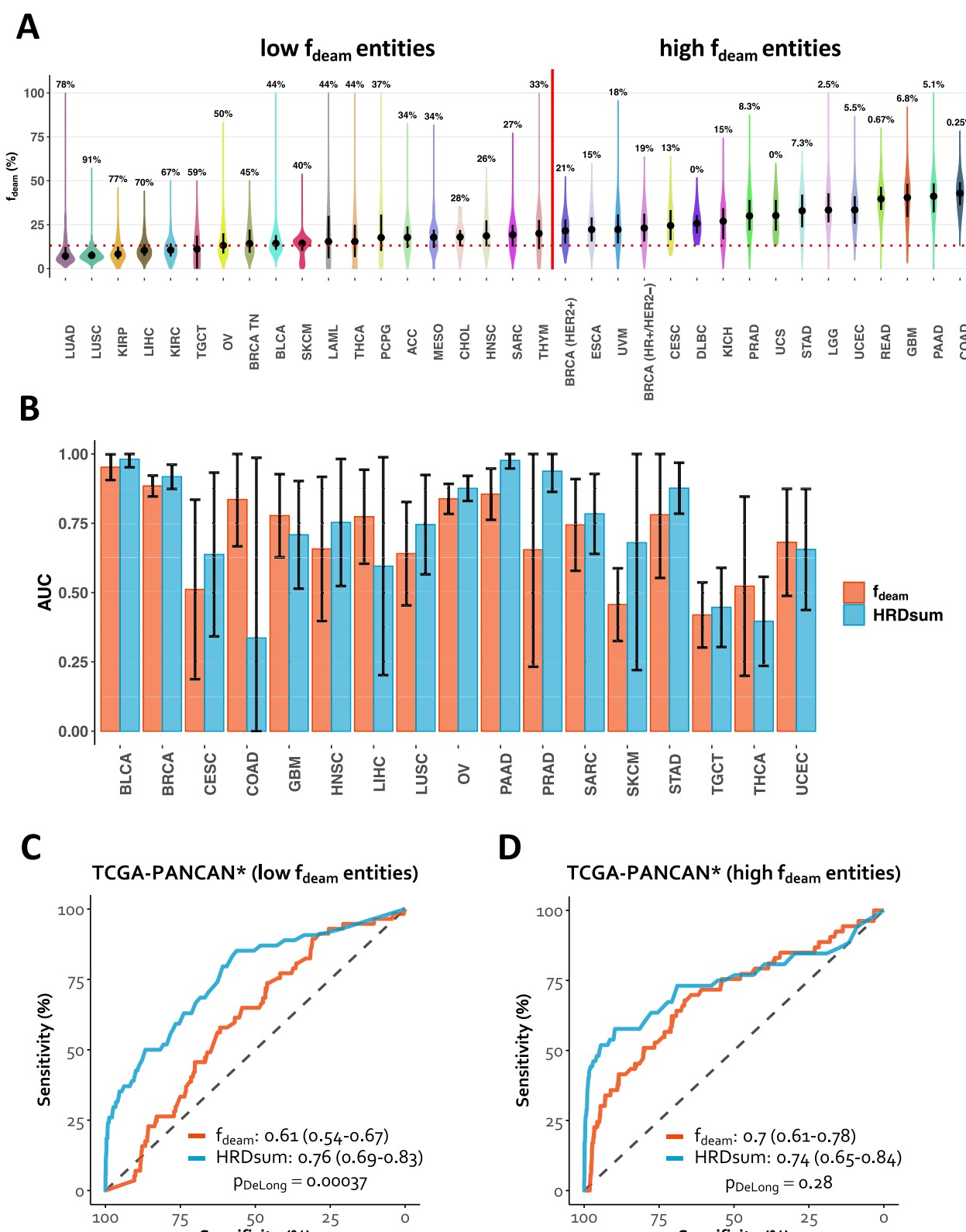

**Fig. 4 | Pan-cancer analysis of the mutations attributed to spontaneous deamination.** Analysis of the fraction of C > T transition at CpG sites relative to all SBSs ($f_{deam}$). **A** Level of $f_{deam}$ across cancer types. Cancer types were stratified into "low $f_{deam}$" and "high $f_{deam}$" based on the percentage of tumors below and the above the cutpoint 13.1% for $f_{deam}$. **B** Performance of $f_{deam}$ was non-inferior to HRDsum for the detection of HRD in 17 cancer types including ovarian and breast cancer. **C, D** Performance of $f_{deam}$ was non-inferior to HRDsum for the detection of HRD in a pooled data set of entities with high $f_{deam}$ ($n = 14$), but inferior to HRDsum in a pooled data set of entities with low $f_{deam}$ ($n = 17$). TCGA-PANCAN* = all entities included in the TCGA except BRCA and OV.

melanoma of the skin (SKCM). For these cancers, the mutational processes operative because of the exposure to these risk factors contribute to lower levels of $f_{deam}$ and can confound the qualification of the biomarker for the detection of HRD.

We analyzed the association of the levels of the new biomarker with a classification of potential causes of HRD (Supplementary Fig. 5). The previously developed classification system distinguishes between tumors with deleterious mutations in *BRCA1/2* (class H1a), tumors with deleterious mutations in other HR-genes (class H1b), tumors with VUS in *BRCA1/2* (class H2a), tumors with VUS in other HR-genes (class H2b), and the remaining tumors (class H3) without deleterious mutations or VUS in HR-genes[13]. If methylation data were available (which was the case for most of the TCGA data sets), tumors with *BRCA1* promoter hypermethylation were included in class H1. For ovarian cancer (in both TCGA-OV and HD-OV), breast cancer (in TCGA-BRCA), and the remaining cancer types (TCGA-PANCAN*), biallelic (BA) mutations of *BRCA1/2* (class H1a) were associated with significantly lower $f_{deam}$ compared with tumors without alterations in the HR-genes (class H3). In all TCGA cohorts (OV, BRCA, and PANCAN*), *BRCA1* hypermethylation was associated with significantly lower $f_{deam}$ compared with tumors in class H3 and the association was similarly strong as the one with deleterious *BRC1/2* mutations. In the ovarian cancer TCGA cohort, additional to class H1a, the classes H1b-MA and H2b were significantly separated from the class H3, but this result could not be validated in HD-OV. In breast cancer (TCGA-BRCA), the levels of $f_{deam}$ were significantly lower for all other classes than for class H3. In the pan-cancer cohort excluding ovarian and breast cancer (TCGA-PANCAN*), additionally to class H1a, classes H1b-MA, H2a, and H2b were significantly separated from class H3. The strong separation of classes H1a-BA and H1a-HM from class H3 by $f_{deam}$ across cancer types is in line with the view that these classes almost exclusively include HR-deficient tumors. For ovarian and breast cancer, but not for the remaining cancer types, a similar strong separation was observed for tumors in class H1a-MA. In some analyses, we observed a weaker but significant separation of classes beyond class H1a from class H3, in line with a putative mixed composition of these classes including both HR-deficient and HR-proficient tumors.

## HRD detection across cancer types

We analyzed the performance of the two biomarkers $f_{deam}$ and HRDsum for the detection of HRD in cancer types beyond ovarian and breast cancer (Fig. 4B–D). First, we separately analyzed 17 cancer types that included at least three tumors with biallelic deleterious *BRCA1/2* mutations or *BRCA1* hypermethylation. The performance of the two biomarkers was not significantly different for those cancer types. Secondly, we grouped the pan-cancer cohort in 17 entities with low $f_{deam}$ and 14 entities with high $f_{deam}$ as shown in Fig 4A. Ovarian and breast cancer were left out from this analysis as they were already analyzed separately. In the analysis of cancer types with typically low $f_{deam}$, HRDsum significantly outperformed $f_{deam}$ (AUC = 0.76 vs. AUC = 0.61, $p = 0.00037$, while the performance of the two biomarkers was not significantly different with typically high $f_{deam}$ entities (AUC = 0.74 vs. AUC = 0.7, $p = 0.28$). The cancer types with typically high $f_{deam}$ included pancreatic adenocarcinoma (PAAD), prostate adenocarcinoma (PRAD), cervical carcinoma (CESC), stomach carcinoma (STAD), endometrial carcinoma (UCEC), colorectal carcinoma (COAD and READ), and others. Thus, the new SBS-based biomarker performed equally well than HRDsum in the detection of HRD across ovarian carcinoma, breast carcinoma, and additional 15 cancer types.

We analyzed the 86 COSMIC MutSigs for the contribution of the mutations defining $f_{deam}$ (Supplementary Table 3). The MutSigs including more than 8% of such mutations were: SBS1 – clocklike signature (89%), SBS6 – defective DNA mismatch repair (47%), SBS10b – POLE mutations (45%), SBS15 – defective DNA mismatch repair (40%), SBS87 – thiopurine chemotherapy (39%), SBS98 – unknown (14%), SBS7a/b – UV light (8%/8%), and SBS44 – DNA mismatch repair (8%). These results explain that $f_{deam}$ cannot be used as biomarker for the detection of HRD in colorectal cancer, stomach cancer, and endometrial cancer in which defective DNA

mismatch repair and defective proofreading are considerable prevalent. This result extends to melanoma as tumor entity associated with UV light exposure and activity of SBS7a/b.

In a pan-cancer analysis, $f_{deam}$ correlated significantly negatively with tumor mutational burden (TMB), while HRDsum correlated significantly positively with TMB (R = −0.23 and R = 0.42, respectively). The same direction of correlations was observed within the majority of cancer types (Supplementary Fig. 6): Of 33 cancer types, significant negative correlations of $f_{deam}$ with TMB were observed for 17 cancer types, while significant positive correlations of HRDsum with TMB were observed for 19 cancer types. In particular, the correlations of $f_{deam}$ with TMB in ovarian and breast cancer were significantly negative (R = −0.45 and R = −0.47), while the correlations of HRDsum and TMB were significantly positive (R = 0.44 and R = 0.55). Given that CG>TG mutations are predominantly driven by the clock-like mutational process underlying SBS1 (Supplementary Table 3), the negative correlations of $f_{deam}$ and TMB are in line with a stronger contribution of other mutational signatures in tumors with high TMB. In particular, it is possible that different intensities or operation times of the mutational process underlying the HRD-associated mutational signature SBS3 in different tumors contribute to the negative correlation, as SBS3 is depleted for CG>TG mutations.

We also compared the classification into HRD-deficient and -proficient tumors using $f_{deam}$ with a cutpoint 13.1% and HRDsum with a cutpoint 42 (Supplementary Fig. 7). A significant agreement of the two biomarkers was observed in ovarian cancer, in the three molecular subtypes of breast cancer, in lung adenocarcinoma, in bladder cancer, in head-and-neck cancer, in stomach cancer, and in endometrial cancer. The inter-test agreement was moderate for ovarian cancer (TCGA-OV: kappa = 0.44, HD-OV: kappa = 0.37) and breast cancer (TCGA-BRCA: kappa = 0.33) but lower for the remaining entities (TCGA-PANCAN*, high $f_{deam}$ entities: kappa = 0.12, TCGA-PANCAN*, low $f_{deam}$ entities: kappa = 0.09). We also performed a cancer type-specific optimization of cutpoints by maximizing the balanced accuracy (Supplementary Table 2). Optimized $f_{deam}$ cutpoints for OV, BRCA, BLCA, STAD, UCEC, and SARC were similar (all between 11% and 19%), while optimized cutpoints estimated for COAD, PAAD, and GBM were higher (between 32% and 39%).

Furthermore, we analyzed the influence of *TP53* mutations on the predictivity of new biomarker (Supplementary Fig. 8). The percentages of H1a* cases classified as $f_{deam}$ low were similar in TP53mt and TP53wt ovarian cancer (TCGA-OV: 88% and 87%) and breast cancer (TCGA-BRCA: both 77%). The statement also applied to H3 cases (TCGA-OV: 20% and 21%; TCGA-BRCA: 11% and 14%) and the remaining H classes. Thus, $f_{deam}$ with cutpoint 13.1% can be used for the prediction of HRD in TP53mt and TP53wt cancers.

## Impact of low tumor purity

To evaluate tumor purity as confounding factor for the detection of HRD, we started with a subcohort of 20 HRD-deficient and 20 HRD-proficient TCGA-OV samples with tumor purity ≥80% and simulated corresponding cohorts of homogenous tumor purity of 60%, 40%, 20%, 10%, and 5% (Fig. 5, Supplementary Fig. 9). Simulation of lower tumor purity was achieved by mixing reads from the tumor tissue with reads from the corresponding normal tissue. Comparing the AUC, the performance of $f_{deam}$ and HRDsum were similar in the cohorts with tumor purity of 80% and 60%, while $f_{deam}$ significantly outperformed HRDsum when the tumors purity was 40%, 20%, and 10%. For tumor purities of 60%, 40%, 20%, 10%, and 5% the balanced accuracy for the detection of HRD with $f_{deam}$ and the predefined cutpoint 13.1% was 90%, 84%, 92%, 82%, and 74%, while the accuracy for the detection of HRD with the predefined cutpoint 42 was 88%, 74%, 75%, 55%, and 50%. In the simulation cohort, the genomic regions sequenced with depth of at least 50×, 100×, and 200× had a mean size of 44 Mb, 28 Mb, and 14 Mb, respectively. Low seqencing coverage decreases the sensitivity for the detection especially of variants with low variant allel fraction explaining the poorer performace of $f_{deam}$ when the tumor purity was very low (5%).

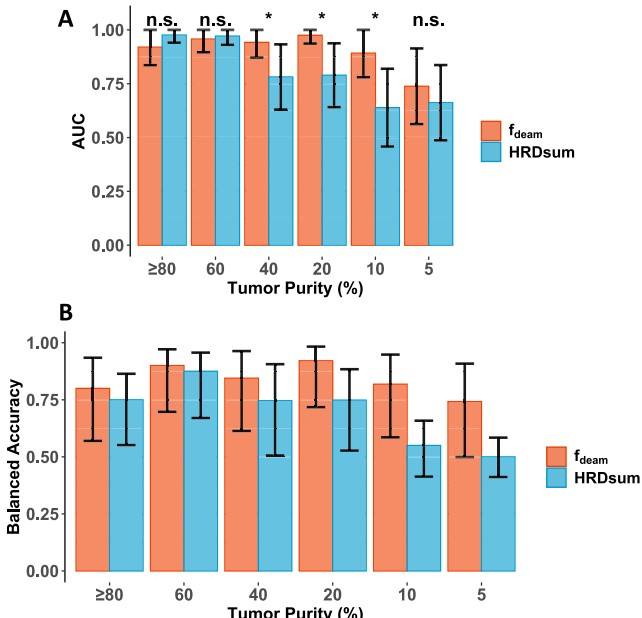

**Fig. 5 | HRD detection in tumors of low tumor purity.** Starting from a cohort of 40 ovarian carcinoma with tumor purity ≥80%, corresponding cohorts with homogenous tumor purity of 60%, 40%, 20%, 10%, and 5% were simulated. **A** ROC analysis of the performance of $f_{deam}$ and HRDsum. **B** Accuracy for the detection of HRD with $f_{deam}$ (cutpoint: 13.1%) and with HRDsum (cutpoint: 42).

## Machine learning

We trained and evaluated classifiers to separate HR-deficient and -proficient tumors (Fig. 6). Two separate series of models were trained, one using ovarian carcinoma as training set and another using pan-cancer as training set. Classifier training included hyperparameter tuning in leave-one-out cross-validation (LOOCV) and subsequent classifier fitting in the training data set. The balanced accuracy obtained was compared between classifiers learned from HRDsum and $f_{deam}$ and classifiers including the 96 mutation types as features. For the model series trained in ovarian carcinoma, a SVM including $f_{deam}$ and HRDsum performed best in both the training set and in the test sets. But the performance of the model including $f_{deam}$ and HRDsum compared with single-feature models including $f_{deam}$ or HRDsum was not significantly better. Also, a SVM including the 96 mutation types instead of $f_{deam}$ and HRDsum did not improve the model performance with the exception of a slight improvement in TCGA-PANCAN*.

The results for the model series trained across cancer types were similar: A SVM including $f_{deam}$ and HRDsum performed best in most test data sets except for TCGA-PANCAN*, in which a SVM including the 96 mutation types and HRDsum performed slightly better. Again, the performance of the 2-feature model compared with single-feature models including $f_{deam}$ or HRDsum was not significantly better. Altogether, balanced accuracies of over 80% were reached for ovarian and breast cancer, while the balanced accuracies for the pan-cancer data set without ovarian and breast cancer did not reach 70%. For ovarian and breast cancer, single-feature classifiers, including $f_{deam}$ and HRDsum, performed almost as good as more complex models.

## Discussion

C > T transitions at CpG sites are the characteristics of signature SBS1 that has been attributed to the clock-like mutational process of spontaneous deamination of methycytosine. Unlike signature SBS1, the mutations caused by HRD are more broadly distributed among mutation types and not dominated by CG > TG transitions building the rational to use $f_{deam}$ for the detection of HRD. In ROC analyses of three cohorts of ovarian cancer and a cohort of breast cancer, $f_{deam}$ was non-inferior to HRDsum in the detection of HRD. The new biomarker had the advantage that it was more robust

against low tumor purity and outperformed HRDsum in detection of HRD in tissue samples with a tumor purity of 40% or less. As shown in the current and an earlier study[13], the sensitivity for the detection of copy number alterations and the accuracy of HRD detection using HRDsum is stepwise decreasing when the tumor purity is 40%, 20% or 10%. By contrast, $f_{deam}$ had a nearly unchanged sensitivity as long as the tumor purity was at least 10%. Thus, the new biomarker may overcome an important diagnostic problem, as acquiring tissue samples of high tumor purity can be difficult or impossible. Especially in neoadjuvant or other settings in which no surgical samples but only biopsies with low tumor cell content are available. Bioinformatically, $f_{deam}$ is easier to implement than HRDsum, it can be directly calculated from the somatic mutation calls, while HRDsum requires determination of the allele-specific CNAs and classification of the resulting genomic scares.

In the current study, we strictly separated between (1) the genetic and epigenetic alterations causing HRD, (2) the effect of HRD to alter the copy numbers in the tumor genome, and (3) the effect of HRD to introduce SBS into the tumor genome. This separation facilitated the comparison of GIS of type (2) and (3) in which the HRD classification (1) served as a reference and guaranteed avoidance of circular reasoning in the comparison of HRD detection methods. Alterations putatively causing HRD were classified into the classes H1a, H1b, H2a, H2b, and H3 based on a classification scheme introduced before[13]. Then, GIS were analyzed for the ability to separate class H1a* including tumors with biallelic deleterious *BRCA1/2* mutations or *BRCA1* hypermethylation from the class H3 including the tumors without genetic alterations in the HR pathway.

A cutpoint of $f_{deam}$ = 13.1% for the new biomarkers was found to maximize the balanced accuracy for calling HRD in TCGA-OV. Using this optimal cutpoint, a sensitivity of 87% and a specificity of 79% was obtained in the training cohort TCGA-OV. In the independent test cohort HD-OV, the cutpoint could be confirmed and achieved a sensitivity of 89% and a specificity of 77%. With comparison for HRDsum with a cutpoint 42, the corresponding values for sensitivity and specificity were 98% and 52% (TCGA-OV) as well as 95% and 47% (HD-OV). The higher sensitivity but lower specificity achieved for HRDsum in comparison with $f_{deam}$ reflects the usage of a cutpoint of 42 for HRDsum which is much lower than the cutpoint of 53.5 that would result from maximizing the balanced accuracy in TCGA-OV. In the breast cancer cohort TCGA-BRCA, a similar optimal cutpoint of 14.9% was found. For other cancer types, some of the optimal cutpoints were higher, but warrant confirmation in further studies, because of the limited numbers of HR-deficient tumors in the TCGA for those cancer types. Studies in cohorts with patients annotated for benefit from PARPi and platinum-based chemotherapy are warranted to further analyze the influence of the cutpoint for $f_{deam}$.

Two different factors of tumor heterogeneity are relevant for the measurement of complex biomarkers: (1) genetic heterogeneity among tumor cells including potential subclonal mutations and CNAs and (2) heterogeneity of the TME with influence on tumor purity. Lower tumor purity results in a lower detection sensitivity for mutations and CNAs, especially for subclonal alterations, and lower $f_{deam}$ and HRDsum scores in turn. Investigating the effect of tumor purity by subgroup analysis of real-world data would require large sample sizes for a well-powered biomarker analyses within each tumor purity bin. In the TCGA, such a subgroup analysis is not feasible, because the project samples were preselected for high tumor purity and most entities do not include samples of low tumor purity. As alternative approach, simulations were employed to analyze the influence of tumor purity on HRD detection. In detail, we started from tumors with high tumor purity and artificially reduced tumor purity by mixing reads from the tumor sample with reads from the surrounding normal tissue. We believe that this method is an excellent model for analyzing the effect of tumor purity on NGS results. The simulations showed that $f_{deam}$ was more sensitive than HRDsum for the detection of HRD in samples of low tumor purity.

The ability of SBS3 and other mutational signatures to separate between HR-deficient and –proficient tumors in ovarian and breast cancers

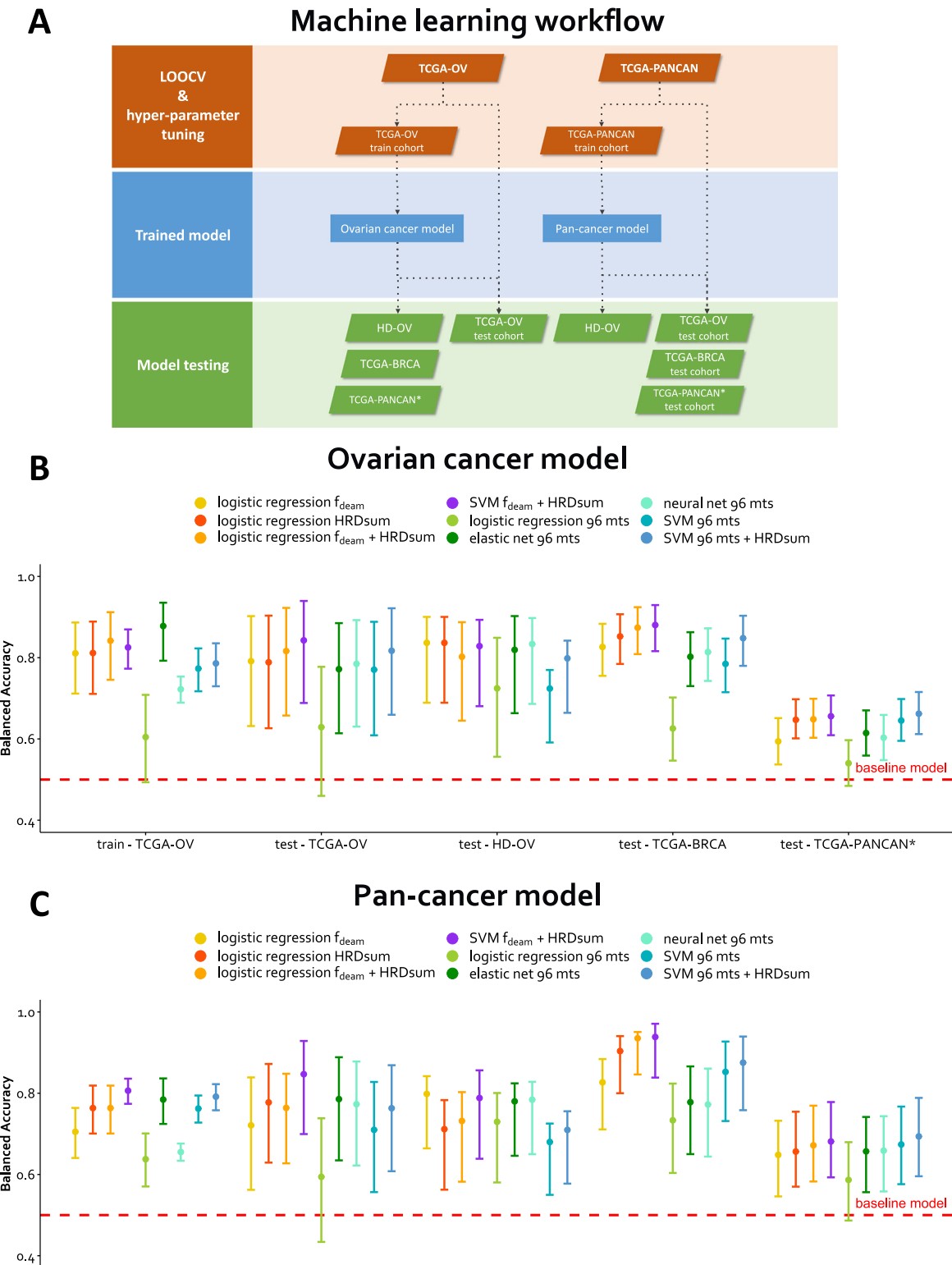

**Fig. 6 | Machine learning to refine the detection of HRD using SBS mutations.** All models were trained in balanced training sets. Training sets were randomly drawn and included 70% of the HR-deficient (class H1a*) tumors and an equal number of HR-proficient tumors (class H3). The remaining tumors were assigned to the test sets. **A** Overview on the data sets and the workflow of cross-validation, model training, and model testing. Two series of models were evaluated: One trained in ovarian cancer and another one trained across cancer types. **B** Performance of the models trained in ovarian cancer. **C** Performance of the models trained in the pan-cancer cohort. TCGA-PANCAN = TCGA pan-cancer cohort including more than 10.000 tumors, TCGA-PANCAN* = TCGA-PANCAN excluding TCGA-BRCA and -OV. HD-OV = in-house ovarian cancer cohort. Mts = mutation types.

was limited and below clinical utility. The tumors included in the study were analyzed by WES and thus the data made up only ~1% of the input for fitting mutational signatures compared to whole genome sequencing (WGS) data. While WGS data are the basis of most published catalogs of SBS signatures[20,21], fitting of mutational signatures based on targeted sequencing data is known to reduce the power for signature detection[22]. Comparing the tools SigProfiler and FitMS for the fitting of mutational signatures, we found that latter method based on two-pass fitting with initial inclusion of the common mutational signatures and later inclusion of the rare mutational signatures provided higher power for the detection of SBS3 in the investigated WES data than the former tool. SigMA is a tool designed for the sensitive detection of the HRD-associated SBS3 in samples with low mutation counts or from targeted gene[23]. While the performance of SigMA in the simulation of WES and panel sequencing data from WGS data was promising, the tool needs additional confirmation in targeted sequencing data from clinical samples. To bypass the limitation of an inaccurate mutational signature fit, we (1) analyzed the ad hoc defined new biomarker $f_{deam}$ and (2) applied machine learning to the 96 mutation types from the basis of the SBS mutational signatures. Based on the data analyzed here, the signatures derived by machine learning models did not significantly outperform $f_{deam}$ in the detection of HRD in ovarian cancer, breast cancer, and other cancer types. Thus, $f_{deam}$ as calculated directly from the classification of SBS detected in a tumor is the biomarker that should be further analyzed for clinical implementation.

The effect of BRCA1/2 loss on mutagenesis has been investigated in model systems of isogenic cell lines[24,25]. In these experiments, a strongly increased mutation rate was observed in homozygous $BRCA1^{-/-}$ and $BRCA2^{-/-}$ cells, but not in heterozygous $BRCA1^{+/-}$ and $BRCA2^{+/-}$ cells. In line with these observations, the new biomarker $f_{deam}$ was numerically (significantly in TCGA-PANCAN*) higher in monoallelic BRCA1/2-mutated tumors compared with biallelic tumors. In the pan-cancer cohort (TCGA-PANCAN*) and the in-house ovarian cancer cohort, the level of $f_{deam}$ was not significantly different between monoallelic BRCA1/2-mutated tumors and tumor without alteration in the HR pathway. These data support the view that the allelic status of BRCA1/2 mutations matter for the development of HRD. From the diagnostic point of view, separation of bi- and monoallelic mutations is aggravated by the imperfect estimation of tumor cell content and partly low tumor purity of clinical samples. In this context, the measurement of genomic biomarkers such as $f_{deam}$ and HRDsum additionally to mutation analysis can help to support HRD calling. In line with the definition of $f_{deam}$, all mutation triples were significantly higher in biallelic BRCA1/2-mutated cells compared to wt cells with the only exception of C > T transitions at CpG sites that did not differ between these cell types[24]. Differences between the patterns of SBS associated with BRCA1 and BRCA2 mutations were neither significant in the published cell culture experiments nor in the tumors analyzed in the current study (Supplementary Fig. 10). Altogether, the observations from the model systems are in line with current study and the suitability of $f_{deam}$ as biomarker for HRD.

The current study focused on an in-depth analysis of the association of patterns of SBSs with HRD using WES data. Other studies have addressed HRD detection at WGS resolution and resulted in the development of the classifiers HRDetect and CHORD[26,27]. HRDetect, a weighted combination of deletions with microhomology, SBS3, SBS8, HRDsum, and two rearrangement signatures was trained and validated in breast cancer, significantly lost power when evaluated on WES data[22]. CHORD, a random forest classifier including mutation and CNA features, was trained in a pancancer cohort and relies on WGS data. Interestingly, C > T transitions were one of six genomic features combined in HRProfiler, a tool for the detection HRD developed in WGS data of breast cancer and presented in a recent preprint[28]. Thus, the development of methods for HRD detection of from NGS data is ongoing and the best algorithm for clinical HRD calling still needs to be defined depending on the type of the sequencing approach. WES offers the opportunity to cover the entire coding regions of the genome, while keeping the region that needs to be sequenced by a factor of 100 smaller in comparison with WGS. The new biomarker $f_{deam}$ can help to additionally read out the HRD status from WES data in a way that is robust against low tumor purity.

A limitation of the new biomarker is its imprecision for tumors of very low TMB (Supplementary Fig. 11). Defined as the percentage of CG > TG mutations, $f_{deam}$ carries a stochastic error that decreases with the inverse square root of the TMB (the latter defined as the total number of detected SBSs). For most of the cancer types, only a few tumors will be affected by large stochastic errors, as more than 90% of tumors had a TMB ≥ 10 mutations in 26 out of the 33 investigated cancer types, including the two most important use cases: ovarian cancer (97.6% in TCGA-OV) and breast cancer (98.5% in TCGA-BRCA). For a clinical implementation of $f_{deam}$, tumors with very low TMB should be called NA, while the precision of $f_{deam}$ for tumors of low TMB could be enhanced by using broader sequencing approaches (e.g., WGS). Regardless of this critical discussion, the study supports the view that $f_{deam}$ is well-suited for HRD calling in ovarian and breast cancer, as only a very small percentage of the tumors of these cancer types have very low TMB, and the demonstrated non-inferiority to HRDsum holds for the entire cohorts of ovarian and breast cancer.

The most important limitations of the current study are the small number of HR-deficient tumors included in the cancer types beyond ovarian and breast cancer. These numbers limit the precise validation of the GIS in these cancer types including a cancer type-specific cutpoint optimization. Additionally, these sample size limitations could be the reason for the limited performance of the biomarkers obtained by machine learning for these cancer types. This limitation could only be overcome by the investigation of exceptionally large cohorts of the cancer types in which HRD is rare. In the simulation study, we observed a decrease of the classification accuracy for extremely low tumor purity such as 5%. Analysis of coverage data suggested that this was due to insufficient sequencing depth and an incomplete mutation calling in turn. We expect that sequencing deeper would enable robust HRD classification of tumors with extremely low tumor purity. Along this line, evaluation of $f_{deam}$ in liquid biopsies is an additional interesting opportunity.

There is a clear trend in precision medicine towards more comprehensive sequencing especially for patients for which all approved treatment lines have been exhausted and national WES and WGS programs have been set up in The Netherlands, United Kingdom, Germany, and other countries[29–31]. Compared to WGS, WES offers the opportunity to extract a large amount of the treatment-relevant information at considerably lower costs[32]. Analyzing WES data, we performed an in-depth analysis of the spectrum of SBS and defined a new biomarker $f_{deam}$ which exploits the reduction of the excess of CG > TG transitions in HR-deficient tumors. The new biomarker performed comparable with HRDsum in the detection of HRD in ovarian and breast cancer and outperformed HRDsum in samples of low tumor purity. In the absence of a gold standard for clinical HRD determination, the new biomarker offers the opportunity of an alternative HRD calling orthogonal to HRD-causing mutations. The clinical validity and usability of $f_{deam}$ and the proposed cutpoint should be further validated, especially in cohorts of ovarian and breast cancer annotated for the benefit from PARPi and in cancer types beyond ovarian and breast cancer. Additionally, the new biomarkers might offer the opportunity for a sensitive evaluation of liquid biopsies for HRD.

## Methods
### Study cohorts
The in-house study cohort comprised 231 ovarian carcinoma samples diagnosed at the Institute of Pathology at the Heidelberg University Hospital (HD-OV) (Supplementary Table 1). Retrospective whole exome sequencing (WES) and data analysis were performed in line with the Declaration of Helsinki and the guidelines of the Ethics Committee of the Medical Faculty at the University of Heidelberg (ethics vote S-315/2020). Mutations were called using the DRAGEN somatic pipeline 4.3.6 (Illumina Inc., San Diego, CA). HRDsum scores were determined as implemented earlier[17]. In short, allele-specific copy numbers (CNs) were

obtained from the WES data of paired tumor and normal tissue samples using Sequenza[33]. Then, LOH, LST, and TAI scores were determined and added up to HRDsum using scarHRD[34]. For the in-house cohort, methylation data were not available.

The Cancer Genome Atlas (TCGA) cohort included 10,199 primary tumors of 33 solid cancer types (TCGA-PANCAN). Among these, there were 411 ovarian (TCGA-OV) tumors and 1026 breast (TCGA-BRCA) tumors, resulting in a pan-cancer cohort of 8762 tumors excluding OV and BRCA (TCGA-PANCAN*) (Supplementary Table 1). Mutation calls (from WES) and methylation data were downloaded from the pan-cancer web page of the GDC Data Portal (https://gdc.cancer.gov/about-data/publications/pancanatlas). HRDsum scores were determined as described earlier[13]. Allele-specific CNs calculated from SNP array data using ASCAT that were obtained from the GDC Data Portal[35]. Then, HRD scores, LOH, LST, TAI, and HRDsum were calculated using scarHRD[34].

Additionally, we analyzed WES data of 68 ovarian carcinoma samples profiled by the Clinical Proteomic Tumor Analysis Consortium cohort (CPTAC-OV) (Supplementary Table 1). BAM files were downloaded from the from GDC Data Portal (https://gdc.cancer.gov/about-gdc/contributed-genomic-data-cancer-research/clinical-proteomic-tumor-analysis-consortium-cptac). Mutations were called using the DRAGEN somatic pipeline 4.3.6 (Illumina Inc., San Diego, CA) and HRDsum scores were determined as implemented earlier[17].

## Genetic and epigenetic status

Tumors were classified according to mutations in genes of the HR pathway using our previously developed classification scheme[13]. Each tumor was assigned to the highest class of the detected mutations: deleterious alteration in *BRCA1/2* (class H1a), deleterious alteration in one of 142 other genes in the HR pathway (class H1b), variant of unknown significance (VUS) in *BRCA1/2* (class H2a), VUS in another gene in the HR pathway (class H2b), and none of these (class H3). Additionally, the deleterious mutations were classified as either biallelic (BA) or monoallelic (MA) and tumors were called for hypermethylation of the *BRCA1* promoter (HM) as described in Rempel et al.[13]. The performance of genomic biomarkers for HRD detection was analyzed comparing tumors with BA mutations of class H1a or *BRCA1* hypermethylation (class H1a*) and tumors without genetic alterations in the HR pathway (class H3).

## Mutational signature fitting

Single Base Substitution (SBS) mutational signatures (MutSigs) were extracted from the spectrum of mutations called from the WES data of each tumor. For this, we compared the performances of the algorithms FitMS developed in the Nik-Zainal lab and SigProfiler developed in the Alexandrov lab[19,20]. For FitMS we used the implementation as R package (signature.tools.lib v2.4.4) and (i) initially did not specify the organ of the tumors (organ = "Other") and (ii) subsequently calculated the signatures organ-specifically for the respective cancer types, e.g., "Ovar" for ovarian cancers and "Breast" for breast cancers. For SigProfiler we used the function *cosmic_fit* from the R package SigProfilerAssignmentR (v0.0.23) together with the Catalogue Of Somatic Mutations In Cancer (COSMICv3.3) reference signatures and (i) initially included all signatures and (ii) subsequently performed new calculations excluding the artifact signatures (27, 43, 45, 46, 47, 48, 49, 50, 51, 52, 53, 54, 55, 56, 57, 58, 59, 60, 95).

## Biomarker definition

SBSs detected by WES were classified according to six possible transitions or transversions in the context of the base in the genomics sequence before and after the mutation resulting in 96 mutation types as described in Alexandrov et al.[36]. Based on this mutation classification, $f_{deam}$ is defined as proportion of C > T transition at CpG sites (mutations of type CG > TG) in relation to all SBSs. These mutations can be attributed to spontaneous deamination of methylcytosine and are a characteristic of the clock-like and age-related mutational signature SBS1[37]. Mutational processes operative beyond methylcytosine deamination are expected to result in a reduction of $f_{deam}$.

Tumor mutational burden (TMB) was defined as the number of SBSs in the genomic region covered by WES. Correlations of the HRD markers and TMB were assessed using Spearman's R.

## Simulation of low tumor purity

We started from a randomly selected subcohort of TCGA-OV including 20 HR-deficient (class H1*) and 20 HR-proficient (class H3) tumors with tumor purity ≥80%. The genomic region covered by sequencing depth of at least 50×, 100×, and 200× in this cohort were 44 ± 13 Mb, 28 ± 10 Mb, and 14 ± 7 Mb (mean ± sd). We used the estimates of tumor purity obtained by pathologists from the histopathological slides and reported by the TCGA. Corresponding cohorts of 40 tumors with homogeneous tumor purity of 60%, 40%, 20%, 10%, and 5% were simulated by subsampling and combining reads from the tumor and the normal BAM files. In brief, when a tumor originally had tumor purity $q$ and the target tumor purity was $p$, a simulated tumor sample was compiled by combining a proportion of $p/q$ reads from the tumor and a proportion of $(1-p/q)$ reads from the normal BAM file.

## Machine learning

Machine learning models for HRD detection were trained to distinguish between HR-deficient (class H1a*) and HR-proficient (class H3) tumors. Two separate series of models were trained, one using ovarian carcinoma as training set and another using tumors across cancer types as training set (Fig. 6A). The training sets were balanced between HR-deficient and -proficient tumors and included 70% of the HR-deficient tumors of TCGA-OV and TCGA-PANCAN, respectively, with samples randomly selected. The remaining samples were used for model testing.

We trained five model types with different predictors. Using the *trainControl* and *train* functions from the R package caret[38], we selected the model type and applied leave-one-out cross-validation (LOOCV). Predictors included 96 mutation types (mts) analyzed via logistic regression (method = "glm"), elastic net, neural network (method = "nnet"), and support vector machines with radial basis function kernel (SVM, method = "svmRadial"). For elastic net, we used the *cv.glmnet* (nlambda = 1000, alpha = 0.5) and *glmnet* from the R package glmnet[39,40]. Logistic regression was also applied to the predictors $f_{deam}$, HRDsum, and $f_{deam}$ + HRDsum. Additionally, the predictors $f_{deam}$ + HRDsum and 96 mts + HRDsum were used for the SVM radial model. Overall, this resulted in 30 different prediction models to test.

## Statistical analysis and visualization

Statistical analyses were performed using R v4.1.2 and RStudio Desktop v2.0.443[41]. Figures were generated using the conventional graphics of the R base package and ggplot2 v3.4.4[42].

Detections of mutational signatures derived from FitMS and SigProfiler in HRD-positive (HR-deficient) and non-HRD (HR-proficient) tumors were evaluated using two-sided Wilcoxon test. We distinguished between the significance levels $p \geq 0.05$ (n.s. = not significant), $0.01 < p < 0.05$ (*), $0.001 < p < 0.01$ (**), $0.0001 < p < 0.001$ (***), and $p < 0.0001$ (****).

Differences in the spectrum of the 96 mutation types between HR-deficient and -proficient tumors were analyzed after normalizing the count of each mutation type by the total number of SBSs. Tumors were classified into HRD-positive and non-HRD groups. The difference in means between the two tumor groups ($\Delta$) was calculated and assessed for significance using the two-sided Wilcoxon test. P-values were corrected for multiple hypothesis testing using the Benjamini–Hochberg (BH) method, and mutation types were compiled controlling the false discovery rate (FDR) at 5%.

The ability of MutSigs, $f_{deam}$, and HRDsum to separate HR-deficient from HR-proficient tumors was analyzed using receiver operating characteristic (ROC) curves as implemented in the R package pROC v1.18.0[43]. Differences in the performance of different biomarkers were assessed using the paired DeLong's test, optimal cutpoints were obtained by maximizing the balanced accuracy. Correlations between $f_{deam}$ and HRDsum were assessed using the Spearman correlation coefficient. Fold changes (FCs)

between HR-deficient and -proficient tumors were assessed for significance using the two-sided Wilcoxon test.

## Reporting summary

Further information on research design is available in the Nature Portfolio Reporting Summary linked to this article.

## Data availability

The TCGA data analyzed in this study can be downloaded from the GDC data portal (https://portal.gdc.cancer.gov/) and the pan-cancer atlas (https://gdc.cancer.gov/about-data/publications/pancanatlas). Somatic variant and copy number alteration (CNA) calls from the HD-OV cohort are available in Supplementary Data 1. The processed data used to generate the figures are provided in Supplementary Data 2.

## Code availability

The R code for statistical analysis and generation of the figures is available upon request from the authors.

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

## Author contributions

E.R.: software, validation, formal analysis, investigation, data curation, writing – original draft. M.M.: software, investigation, writing – review & editing. K.K.: software, investigation, writing – review & editing. S.B.: software, investigation, writing – review & editing. M.B.: writing – review & editing. P.S.: resources, writing – review & editing. D.K.: resources, writing – review & editing. A.S.: resources, writing – review & editing. J.B.: conceptualization, methodology, validation, investigation, writing – original draft, supervision, project administration.

## Funding

## Competing interests

P.S. reports personal fees for speaker honoraria BMS, grants from BMS, AstraZeneca, MSD, and boards from BMS, AstraZeneca, MSD, outside the submitted work. D.K. reports personal fees for speaker's honoraria from AstraZeneca, and Pfizer, personal fees for Advisory Board from Bristol-Myers Squibb, outside the submitted work. A.S. has received advisory boards from Agilent, Aignostics, Amgen, Astellas, Astra Zeneca, Bayer, BMS, Eli Lilly, Illumina, Incyte, Janssen, MSD, Novartis, Pfizer, Qlucore, QuiP, Roche, Sanofi, Seagen, Servier, Takeda, Thermo Fisher, and grants from Bayer, BMS, Chugai, Incyte, MSD, outside the submitted work. J.B. reports grants from German Cancer Aid and consulting from MSD, outside the submitted work. All remaining authors declare that they have no conflict of interest, outside the submitted work.
