## [Transparent Peer Review file · Communications Biology]

CG>TG mutation frequency as negative predictor of homologous recombination deficiency in ovarian and breast cancer

Corresponding Author: Professor Jan Budczies

Version 0:

Reviewer comments:

Reviewer #1

(Remarks to the Author)

Manuscript Overview

In the manuscript titled CG>TG Mutation Frequency as a Negative Predictive Indicator for HRD Detection, the authors explored a novel biomarker, fdeam, defined as the CpG>TpG mutation frequency relative to all single base substitutions, for detecting homologous recombination deficiency (HRD). Using data from ovarian and breast cancer cohorts, they demonstrated that fdeam performs comparably to the established HRDsum metric, especially in low tumor purity scenarios. This exploratory study introduces an innovative biomarker with potential clinical utility, particularly in contexts where traditional HRD detection methods face limitations.

While the study is well-conceived and provides valuable insights, several methodological and analytical issues require attention before the manuscript can be considered for publication.

Major Comments

1. Simulation of Tumor Purity

The simulation of low tumor purity is scientifically sound; however, the manuscript does not critically address the potential limitations of this approach, such as the lack of real-world tumor heterogeneity and the exclusion of immune or stromal cell contributions. A more nuanced discussion of these limitations is necessary to contextualize the findings.

Additionally, while the simulation details are outlined in the Methods section, providing a brief summary in the Results section would enhance clarity for readers.

2. Validation of fdeam Across Cancer Types

The study primarily focuses on ovarian and breast cancer, with limited validation in other cancer types. The manuscript should discuss whether the proposed threshold for fdeam (13.1%) is robust across diverse tumor contexts or if it requires optimization for specific cancers.

3. Dependence on Public Databases

The reliance on TCGA and HD-OV datasets is appropriate but introduces potential biases, as these datasets may not fully exclude patients with other cancer diagnoses or confounding conditions. The manuscript should clarify whether patient cohorts were screened for additional malignancies or significant comorbidities that could affect HRD detection.

Minor Comments

1. Figure 4A: While the manuscript defines "Low fdeam" and "High fdeam" (13.1% threshold), this definition could be reiterated in the figure legend for clarity.

2. Line 708: There is a typo: "correponding" should be corrected to "corresponding."

Reviewer #2

(Remarks to the Author)

This is an interesting study that seeks to identify a new marker of HRD. However, there are some concerns.

1) As they state in the Discussion, a preprint paper (PMID 39040162) reported that HRP tumors, i.e., tumors that are not HRD, in breast and ovarian cancer have a characteristic of C to T transitions at CpG sites.

<https://www.medrxiv.org/content/10.1101/2024.07.14.24310383v1>

The authors should cite PMID 39040162 in the Introduction. They should then rewrite the entire paper based on the existence of PMID 39040162, and clearly state what is new about this study.

2) One difference between this study and PMID 39040162 is that fdeam is being attempted in PANCAN. In addition to HRD, gene mutations for various reasons can be the reason for carcinogenesis. In order for “the low rate of C to T transitions at CpG sites” to be a biomarker for HRD in PANCAN, “C to T transitions at CpG sites” must be a characteristic of all gene mutagens other than HRD. However, the fact that fdeam is not useful in PANCAN indicates that the above (unlikely) hypothesis is wrong.

The authors should identify tumors (or mutagens) that are prone to “C to T transitions at CpG sites” before performing PANCAN analysis, and perform fdeam analysis on such tumors.

3) The relationship between TP53 mutations and genomic scars due to HRD has been shown (PMID 35613413). The analysis should be divided into two groups: with and without TP53 mutations.

4) It should also be verified using other publicly available data (ICGC and CPTAC).

5) Figure 1: This is irrelevant to the main story and should be removed from the main figure.

5) Figure 6: This is a figure of low importance and should be removed from the main figure.

Version 1:

Reviewer comments:

Reviewer #1

(Remarks to the Author)

The revised manuscript addresses all substantive scientific critiques raised in the previous round, and the point-by-point rebuttal demonstrates that each reviewer request has been met with new analyses, text additions or clarified figures. Methodological soundness, novelty of the fdeam biomarker, and the expanded validation in CPTAC-OV are now convincing. I therefore recommend acceptance without further revision.

Reviewer #2

(Remarks to the Author)

The authors responded appropriately to some of the comments. However, there are still the following concerns.

1) The number of gene mutations is low in normal (non-tumor) cells. In that case, will the fdeam score be low? The authors should show the fdeam and HRDsum values for non-tumor samples and indicate whether non-tumor samples are judged to be HRD. Of course, in reality, non-tumor samples are not HRD, so if they are judged to be HRD by fdeam, this should be stated as a limitation of fdeam.

2) Does the fdeam score correlate with TMB? Whether or not fdeam correlates with TMB is thought to depend on which mutational signature the tumor has or which cancer type it is. The authors should analyze this.

Version 2:

Reviewer comments:

Reviewer #2

(Remarks to the Author)

The authors revised the manuscript appropriately.

Re: COMMSBIO-24-7318-T

Point-to-point response to the reviewer comments

Reviewer #1

Manuscript Overview: In the manuscript titled CG>TG Mutation Frequency as a Negative Predictive Indicator for HRD Detection, the authors explored a novel biomarker, f_{deam} , defined as the CpG>TpG mutation frequency relative to all single base substitutions, for detecting homologous recombination deficiency (HRD). Using data from ovarian and breast cancer cohorts, they demonstrated that f_{deam} performs comparably to the established HRDsum metric, especially in low tumor purity scenarios. This exploratory study introduces an innovative biomarker with potential clinical utility, particularly in contexts where traditional HRD detection methods face limitations. While the study is well-conceived and provides valuable insights, several methodological and analytical issues require attention before the manuscript can be considered for publication.

Response: We thank the reviewer for the encouraging summary.

Comment #1: Simulation of Tumor Purity. The simulation of low tumor purity is scientifically sound; however, the manuscript does not critically address the potential limitations of this approach, such as the lack of real-world tumor heterogeneity and the exclusion of immune or stromal cell contributions. A more nuanced discussion of these limitations is necessary to contextualize the findings. Additionally, while the simulation details are outlined in the Methods section, providing a brief summary in the Results section would enhance clarity for readers.

Response: Responding to the comment on tumor heterogeneity, we added the following paragraph to the Discussion section:

Prof. Dr. rer. nat. J. Budczies
Head of Bioinformatics

Institute of Pathology
Department of General Pathology
and Pathological Anatomy

Heidelberg, 26.03.2025

Im Neuenheimer Feld 224
69120 Heidelberg
Tel. +49 6221 56-32757
Fax +49 6221 56-52 51

“Two different factors of tumor heterogeneity are relevant for the measurement of complex biomarkers: (1) genetic heterogeneity among tumor cells including potential subclonal mutations and CNAs and (2) heterogeneity of the TME with influence on tumor purity. Lower tumor purity results in a lower detection sensitivity for mutations and CNAs, especially for subclonal alterations, and lower f_{deam} and HRDsum scores in turn. Investigating the effect of tumor purity by subgroup analysis of real-world data would require large sample sizes for a well-powered biomarker analyses within each tumor purity bin. In the TCGA, such a subgroup analysis is not feasible, because the project samples were preselected for high tumor purity and most entities do not include samples of low tumor purity. As alternative approach, simulations were employed to analyze the influence of tumor purity on HRD detection. In detail, we started from tumors with high tumor purity and artificially reduced the tumor purity by mixing reads from the tumor sample with reads from the surrounding normal tissue. We believe that this method is an excellent model for analyzing the effect of tumor purity on NGS results. The simulations showed that f_{deam} was more sensitive than HRDsum for the detection of HRD in samples of low tumor purity.”

Following the reviewer’s suggestion, we added more details of the simulation methodology to the Results section (lines 259-263):

“To evaluate tumor purity as confounding factor for the detection of HRD, we started with a subcohort of 20 HRD-deficient and 20 HRD-proficient TCGA-OV samples with tumor purity $\geq 80\%$ and simulated corresponding cohorts of homogenous tumor purity of 60%, 40%, 20%, 10%, and 5% (Figure 5, Suppl. Figure 8). Simulation of lower tumor purity was achieved by mixing reads from the tumor tissue with reads from the corresponding normal tissue.”

Comment #2: Validation of f_{deam} Across Cancer Types
The study primarily focuses on ovarian and breast cancer, with limited validation in other cancer types. The manuscript should discuss whether the proposed threshold for f_{deam} (13.1%) is robust across diverse tumor contexts or if it requires optimization for specific cancers.

Response: The original manuscript already included a cancer type specific cutpoint optimization, please see Suppl. Table 2. The results of this analysis are reported in the Results section (lines 246-250):

“We also performed a cancer type-specific optimization of cutpoints by maximizing the balanced accuracy (Suppl. Table 2). Optimized f_{deam} cutpoints for OV, BRCA, BLCA, STAD, UCEC, and SARC were similar (all between 11% and 19%), while optimized cutpoints estimated for COAD, PAAD, and GBM were higher (between 32% and 39%).”

Responding to the reviewer’s comment we added a cutpoint analysis in the two additional ovarian cancer cohorts and obtained optimal cutpoints of 14.9% and 11.0% for the HD-OV and the CPTAC-OV cohort, respectively, similar to the optimal cutpoint of 13.1% obtained in the TCGA-OV cohort.

In summary, while the cutpoint of 13.1% works well for ovarian and breast cancer, optimal cutpoints for other cancer types might be higher and warrant further analysis, because the number of HR-deficient tumors for these cancer types in the TCGA is limited. Responding to the reviewer’s comment, we added the following statement to the Discussion section (lines 336-339):

“In the breast cancer cohort TCGA-BRCA, a similar optimal cutpoint of 14.9% was found. For other cancer types, some of the optimal cutpoints were higher, but warrant confirmation in further studies, because of the limited numbers of HR-deficient tumors in the TCGA for those cancer types.”

Comment #3: Dependence on Public Databases: The reliance on TCGA and HD-OV datasets is appropriate but introduces potential biases, as these datasets may not fully exclude patients with other cancer diagnoses or confounding conditions. The manuscript should clarify whether patient cohorts were screened for additional malignancies or significant comorbidities that could affect HRD detection.

Response: HRD is a cell-specific biological property meaning that the cell is unable to repair DNA double strand breaks by using sequence information from the homologous chromosome. In the current study, we investigated HRD as a property of tissue samples. The classification HR-deficient vs. proficient is expected to apply to the tumor from which the tissue sample

was taken, while no statements are made on other tumors of the same patient. Analysis of multiple tumors or metastases of the same patient and a shared or unshared HRD status is out of the scope of the current study. Comorbidities are not expected to act as confounding factors in the current study, as we are investigating association between genetic properties of the tumor cell in the tissue samples.

Comment #4 (minor concern): Figure 4A: While the manuscript defines "Low f_{deam} " and "High f_{deam} " (13.1% threshold), this definition could be reiterated in the figure legend for clarity.

Response: We expanded the figure legend accordingly.

Comment #4 (minor concern): Line708: There is a typo: "correponding" should be corrected to "corresponding."

Response: We corrected the typo.

Reviewer #2

Reviewer #2: This is an interesting study that seeks to identify a new marker of HRD. However, there are some concerns.

Response: We thank the reviewer for the encouraging summary.

Comment #1: As they state in the Discussion, a preprint paper (PMID 39040162) reported that HRP tumors, i.e., tumors that are not HRD, in breast and ovarian cancer have a characteristic of C to T transitions at CpG sites. <https://www.medrxiv.org/content/10.1101/2024.07.14.24310383v1>. The authors should cite PMID 39040162 in the Introduction. They should then rewrite the entire paper based on the existence of PMID 39040162, and clearly state what is new about this study.

Response: The reviewer is referring to a preprint that did not went through peer review so far. We found the preprint when the analyses for the current manuscript were finished and added it to the Discussion section. The preprint did neither influenced study design, nor statistical analyses, nor the figures produced for the manuscript. The two studies were carried out in parallel and submitted in quick succession (The preprint appeared at July 14th, 2024. The manuscript was finished and submitted to Nature

Communications at October 17th, 2024.). As study design and analyses were finished performed before the preprint appeared, we prefer to cite it in the Discussion and not in the Introduction section.

Additionally, we would like to stress that focus and details of the current study and the preprint (PMID 39040162) are different in many aspects:

- The current study focusses on HRD detection based on the somatic mutations without the inclusion of CNAs, while the preprint focusses on building an HRD classifier integrating somatic mutations and CNAs
- As a very important feature, the classifier based solely on somatic mutations is more sensitive for HRD detection when the tumor purity compared to classifier based on CNA. Limited tumor purity is an important bottleneck for HRD detection in some of the clinical samples. The current manuscript includes an extensive simulation analysis showing that somatic mutation-based classification outperforms CNA-based classification when the tumor purity is low.
- In the current study, a scheme to classify tumors based on genetic and epigenetic alterations is used enabling a direct head-to-head comparison of the genomic biomarkers f_{deam} and HRDsum.
- For ovarian cancer, the current study includes two additional validation data sets HD-OV and CPTAC-OV. The former cohort includes patients treated at the Heidelberg University Hospital and samples processed via our routine diagnostics workflow (WES starting from FFPE samples).
- The current study includes a pan-cancer analysis beyond ovarian and breast cancer in the TCGA cohort

Comment #2: One difference between this study and PMID 39040162 is that f_{deam} is being attempted in PANCAN. In addition to HRD, gene mutations for various reasons can be the reason for carcinogenesis. In order for “the low rate of C to T transitions at CpG sites” to be a biomarker for HRD in PANCAN, “C to T transitions at CpG sites” must be a characteristic of all gene mutagens other than HRD. However, the fact that f_{deam} is not useful in PANCAN indicates that the above (unlikely) hypothesis is wrong. The authors should identify tumors (or mutagens) that are prone to “C to T

transitions at CpG sites” before performing PANCAN analysis, and perform f_{deam} analysis on such tumors.

Response: We thank the reviewer for the interesting comment. The new biomarkers f_{deam} was non-inferior to HRDsum in ovarian and cancer breast cancer. Furthermore, analyzing the TCGA data excluding ovarian and breast cancer, non-inferiority was observed in pooled data from 14 cancer types, while it was not observed in pooled data from the remaining 17 cancer types. We hypothesized that this difference is due to the distinct mutational processes operative in specific cancer types. To analyze this in more detail, we investigated the contribution of CG>TG mutations to each mutational signature in the COSMIC catalog and added the findings to the Results section of the manuscript (lines 230-238):

“We analyzed the 86 COSMIC MutSigs for the contribution of the mutations defining f_{deam} (Suppl. Table 3). The MutSigs including more than 8% of such mutations were: SBS1 – clocklike signature (89%), SBS6 – defective DNA mismatch repair (47%), SBS10b – POLE mutations (45), SBS15 – defective DNA mismatch repair (40%), SBS87 – thiopurine chemotherapy (39%), SBS98 – unknown (14%), SBS7a/b – UV light (8%/8%), and SBS44 – DNA mismatch repair (8%). These results explain that f_{deam} cannot be used as biomarker for the detection of HRD in colorectal cancer, stomach cancer, and endometrial cancer in which defective DNA mismatch repair and defective proofreading are considerable prevalent. This result extends to melanoma as tumor entity associated with UV light exposure and activity of SBS7a/b.”

Comment #3: The relationship between TP53 mutations and genomic scars due to HRD has been shown (PMID 35613413). The analysis should be divided into two groups: with and without TP53 mutations.

Response: Responding to the reviewer comment we preformed a TP53-stratified analysis and added the findings to the Results section of the manuscript (lines 251-256):

“Furthermore, we analyzed the influence of *TP53* mutations on the predictivity of new biomarker (Suppl. Figure 7). The percentages of H1a* cases classified as f_{deam} low were similar in TP53mt and TP53wt ovarian cancer (TCGA-OV: 88% and 87%) and breast cancer (TCGA-BRCA: both 77%). The statement also applied to H3 cases (TCGA-OV: 20% and 21%; TCGA-

BRCA: 11% and 14%) and the remaining H classes. Thus, f_{deam} with cutpoint 13.1% can be used for the prediction of HRD in TP53mt and TP53wt cancers.”

Comment #4: It should also be verified using other publicly available data (ICGC and CPTAC).

Response: So far, the manuscript included the large TCGA pan-cancer cohort and the clinical HD-OV cohort as validation cohort. Currently, ovarian cancer is the only cancer type with HRD scores being part of treatment approvals and thus represents the most important use case. Thus and because of being collected in a real-world clinical setting (patients treated at the Heidelberg University Hospital, FFPE tissue samples), we believe that the HD-OV is an excellent choice for validation. While we still believe that the data as originally presented were sufficient for establishment of the new biomarker, we followed the suggestion of the reviewed and added the analysis of CPTAC-OV as a third ovarian cancer cohort.

The new analysis of the CPTAC-OV cohort confirmed the findings in TCGA-OV and HD-OV. Again, we detected a strong negative and significant correlation between f_{deam} and HRDsum and a non-inferior performance f_{deam} compared to HRDsum in the detection of HRD (Suppl. Figure 4, f_{deam} performed numerically better than HRDsum). The new results were added to the manuscript as follows (lines 173-178 and lines 465-470):

“As additional validation of f_{deam} for the detection of HRD in ovarian cancer, we analyzed the independent cohort CPTAC-OV (Suppl. Figure 4). Again, we detected a strong and significant correlation between f_{deam} and HRDsum (Spearman $R = -0.54$). Also, the mutation-based biomarker was non-inferior to HRDsum to separate between HR deficient and HR-proficient tumors (AUC=0.79 vs. AUC=0.65, $p=0.39$). In summary, f_{deam} was non-inferior to HRDsum for the detection of HRD in three cohorts of ovarian cancer and one cohort of breast cancer.”

“Additionally, we analyzed WES data of 68 ovarian cancers profiled by the Clinical Proteomic Tumor Analysis Consortium cohort (CPTAC-OV) (Suppl. Table 1). BAM files were downloaded from the from GDC Data Portal (<https://gdc.cancer.gov/about-gdc/contributed-genomic-data-cancer-research/clinical-proteomic-tumor-analysis-consortium-cptac>). Mutations were called using the DRAGEN somatic pipeline 4.3.6 (Illumina Inc., San

Diego, CA) and HRDsum scores were determined as implemented earlier [17].”

Comment #5: Figure 1: This is irrelevant to the main story and should be removed from the main figure. Figure 6: This is a figure of low importance and should be removed from the main figure.

Response: We thank the reviewer for the thoughts on the two figures. Figure 1 shows that the sensitivity and specificity for the detection of HRD using COSMIC and FitMS mutational signatures calculated from WES data are limited. Because this result motivates the search for a better SBS-based biomarkers, we would like to keep the figure in the manuscript.

We completely agree with the author that Figure 6 shows as negative result: Machine learning from the set of somatic mutations did not considerably improve the predictivity compared to f_{deam} that was defined *ad hoc*. But we believe that this result is important demonstrating the optimality of f_{deam} compared to other classifiers trained on somatic mutations. Therefore, we would like to keep the figure in the manuscript.

Re: COMMSBIO-24-7318B

Point-to-point response to the reviewer comments

Reviewer #2

The authors responded appropriately to some of the comments. However, there are still the following concerns.

Response: These concerns are new and were not part of the first review round, but we are happy to discuss them.

Reviewer #2, Comment #1: The number of gene mutations is low in normal (non-tumor) cells. In that case, will the f_{deam} score be low? The authors should show the f_{deam} and HRDsum values for non-tumor samples and indicate whether non-tumor samples are judged to be HRD. Of course, in reality, non-tumor samples are not HRD, so if they are judged to be HRD by f_{deam} , this should be stated as a limitation of f_{deam} .

Response: Both f_{deam} and HRDsum aim on the detection of HRD, a property that the cancer cells gain after the malignant transformation and are based on somatic point mutations and somatic copy number alterations, respectively. In the current study, f_{deam} and HRDsum are calculated based WES data of matched tumor and normal DNA and corresponding somatic mutation and somatic copy alteration calls, respectively. Technically, f_{deam} is not calculatable when no somatic mutations are detected. Please see below (Response to Comment #1 and #2) for a more general discussion on f_{deam} for samples with very low TMB.

Recently, there have been investigation on the accumulation of somatic mutations in normal tissues, but these alterations are subclonal and present at very low VAFs. Calling and investigation of these alterations is not feasible in the TCGA and the HD data that were sequenced with limited sequencing

Prof. Dr. rer. nat. J. Budczies
Head of Bioinformatics

Institute of Pathology
Department of General Pathology
and Pathological Anatomy

Heidelberg, 09.05.2025

Im Neuenheimer Feld 224
69120 Heidelberg
Tel. +49 6221 56-32757
Fax +49 6221 56-52 51

depth. Also, this objective is beyond the scope of the study that focuses on the detection of HRD in tumor tissues.

Reviewer #2, Comment #2: Does the f_{deam} score correlate with TMB? Whether or not f_{deam} correlates with TMB is thought to depend on which mutational signature the tumor has or which cancer type it is. The authors should analyze this.

Response: As suggested, we calculated the correlations of f_{deam} and TMB in the TCGA cohort (Suppl. Figure 6A). Additionally, we calculated the correlations of HRDsum und TMB (Suppl. Figure 6B). We added the following paragraph to the Results section (lines 239-252):

“In a pan-cancer analysis, f_{deam} correlated significantly negatively with tumor mutational burden (TMB), while HRDsum correlated significantly positively with TMB ($R=-0.23$ and $R=0.42$, respectively). The same direction of correlations was observed within the majority of cancer types (Suppl. Figure 6): Of 33 cancer types, significant negative correlations of f_{deam} with TMB were observed for 17 cancer types, while significant positive correlations of HRDsum with TMB were observed for 19 cancer types. In particular, the correlations of f_{deam} with TMB in ovarian and breast cancer were significantly negative ($R=-0.45$ and $R=-0.47$), while the correlations of HRDsum and TMB were significantly positive ($R=0.44$ and $R=0.55$). Given that CG>TG mutations are predominantly driven by the clock-like mutational process underlying SBS1 (Suppl. Table 3), the negative correlations of f_{deam} and TMB are in line with a stronger contribution of other mutational signatures in tumors with high TMB. In particular, it is possible that different intensities or operation times of the mutational process underlying the HRD-associated mutational signature SBS3 in different tumors contribute to the negative correlation, as SBS3 is depleted for CG>TG mutations.”

Response to Comments #1 and #2: An important aspect of TMB is the connection is its connection stochastic error of f_{deam} that is defined as the percentage of CG>TG mutations. Stimulated by the reviewer comments, we have now analyzed this dependency in more detail and added the following paragraph to the Discussion section (lines 427-439):

“A limitation of the new biomarker is its imprecision for tumors of very low TMB (Suppl. Figure 11). Defined as the percentage of CG>TG mutations, f_{deam}

carries a stochastic error that decreases with the inverse square root of the TMB (the latter defined as the total number of detected SBS). For most of the cancer types, only a few tumors will be affected by large errors, as more than 90% of the tumors had $TMB \geq 10$ mutations for most of the cancer types (9,529 of 10,199) including the two most important use cases of ovarian cancer (97.6% in TCGA-OV) and breast cancer (98.5% in TCGA-BRCA). For a clinical implementation of f_{deam} , tumors with very low TMB should be called NA, while the precision of f_{deam} for tumors of low TMB could be enhanced by using broader sequencing approaches (e.g., WGS). Regardless of this critical discussion, the study supports the view that f_{deam} is well-suited for HRD calling in ovarian and breast cancer, as only a very small percentage of the tumors of these cancer types have very low TMB, and the demonstrated non-inferiority to HRDsum holds for the entire cohorts of ovarian and breast cancer.”